# The Effects of Individual Characteristics of the Naval Personnel on Sleepiness and Stress during Two Different Watchkeeping Schedules

**DOI:** 10.3390/ijerph192013451

**Published:** 2022-10-18

**Authors:** Mikko Myllylä, Heikki Kyröläinen, Tommi Ojanen, Juha-Petri Ruohola, Olli J. Heinonen, Petteri Simola, Tero Vahlberg, Kai I. Parkkola

**Affiliations:** 1Centre for Military Medicine, The Finnish Defence Forces, 20241 Turku, Finland; 2Doctoral Programme in Clinical Research, University of Turku, 20014 Turku, Finland; 3Neuromuscular Research Center, Faculty of Sport and Health Sciences, University of Jyväskylä, 40014 Jyväskylä, Finland; 4Department of Leadership and Military Pedagogy, National Defence University, 00861 Helsinki, Finland; 5Human Performance Division, Finnish Defence Research Agency, The Finnish Defence Forces, 04310 Tuusula, Finland; 6The Navy Command Finland, The Finnish Defence Forces, 20811 Turku, Finland; 7Paavo Nurmi Centre & Unit of Health and Physical Activity, University of Turku, 20520 Turku, Finland; 8Department of Biostatistics, University of Turku, 20014 Turku, Finland; 9Faculty of Medicine and Health Technology, Tampere University, 33100 Tampere, Finland

**Keywords:** individual characteristics, individual factors, shift work, watchkeeping, irregular working hours, sleepiness, fatigue, stress, navy

## Abstract

Background: Naval service can have a significant impact on the wellbeing of seafarers, and the operation of warships is highly dependent on the personnel on board. Nevertheless, there is a lack of knowledge concerning the impact of seafarers’ individual characteristics on their wellbeing in a naval environment. Therefore, the aim of this study was to investigate individual characteristics of the naval personnel that may be associated with the amount of sleepiness, fatigue and stress responses experienced during shift work and irregular working hours in a naval environment. Methods: The study took place on a Finnish Defence Forces’ Navy missile patrol boat on which 18 crewmembers served as study participants. The measurement periods lasted two separate weeks (seven days and six nights) during shift work with two different watchkeeping systems (4:4, 4:4/6:6). The onboard measurements consisted of the Karolinska Sleepiness Scale, salivary stress hormones, cognitive tests (Sustained Attention to Response Task and N-back Task) and heart rate variability. Results: Participants of older ages or with a longer history in naval service were associated with a greater amount of sleepiness, fatigue and stress responses on board. On the contrary, increased physical activity and a higher level of physical fitness, especially standing long jump, were associated with a lower amount of sleepiness, fatigue and fewer stress responses. In addition, an athletic body composition together with a healthy lifestyle may be beneficial, considering the stress responses on board. Conclusion: The present results are well in line with the previous literature regarding shift work and irregular working hours. The results highlight the importance of regular physical activity and good physical fitness during service in the naval environment.

## 1. Introduction

The operation of naval ships demands a capability which is highly dependent on the personnel on board. The naval environment is quite isolated, and naval service can have a significant impact on the physical and psychological wellbeing of seafarers. Thus, it is important to identify the factors which are associated with the wellbeing of the crew in a naval environment.

In the military, there are usually requirements for the minimum level of physical fitness. Soldiers are generally expected to be physically fit and capable of completing physically demanding tasks. While being physically fit has, in general, been recognized as an important characteristic in the military [1], there is a lack of knowledge concerning the importance of physical fitness in the naval environment. Previous studies have reported the most physically demanding military naval tasks [2,3]. The categories which have been suggested to be physically demanding in the naval environment are: casualty handling, firefighting, damage control and basic transition tasks [3]. Out of these categories, casualty handling has been suggested to be the single most demanding task in terms of muscle strength and endurance [3]. In the case of firefighting on board, a prior study suggested that naval personnel must achieve a maximal oxygen uptake (VO_2_max) value of 41 mL/kg/min as an absolute minimum standard regarding endurance performance [4]. Regarding stress, greater levels of cardiorespiratory and muscular fitness have been associated with lower stress symptoms among normal-weight men [5,6]. Physical activity has also been associated with less subjective psychological stress [7]. However, there is still a lack of knowledge concerning the importance of physical fitness in terms of fatigue and stress responses during shift work and irregular working hours in a naval environment.

Concerning body composition, a higher body mass index (BMI) and body fat percentage (BFP) have been associated with increased levels of stress [8,9]. Obesity is known to increase the risk of certain health problems (e.g., cardiovascular diseases, diabetes, musculoskeletal disorders and some cancers) which can be problematic when they occur on board [10,11]. Obesity is also suggested to increase the danger of being on board because it may be difficult for obese persons to carry out physically demanding tasks in an emergency situation [11]. Limited influence over the served food quality and easy access to large portions of food have been identified as possible explanations contributing to obesity among seafarers [12,13]. A recent study regarding military personnel in the U.S. reported that in age groups 20 years or older there were more personnel in the overweight or obese category in the navy (67.0%) than in the coast guard (66.6%), army (63.2%), air force (62.9%) or marine corps (57.8%) [9]. The percentage of obese or overweight personnel also increased with age [14]. A previous study in the U.S. Navy reported that the relative number of overweight personnel was 69% on small submarines, 66% on large submarines and 63% on aircraft carriers [15]. The study infers that the more confined the vessel, the higher the percentage of overweight crew.

Concerning the psychological aspects of a maritime environment, naval service personnel are dealing with the same kind of stress factors as general seafarers [16]. Working on a merchant ship is a mentally and physically challenging occupation that has the potential to induce even severe psychological distress [17,18,19]. In particular, separation from family, time pressure at work, long working days and heat in working areas are the most stressful factors for seafarers [20]. It has been recognized that an individual’s personality is related to one’s ability to deal with occupational stress [21]. In terms of shift work, extraversion has been positively linked and neuroticism negatively linked with shift work tolerance [22]. It has also been discovered that openness may be a protective factor against burnout [23]. When assessing the objective level of psychological stress, salivary alfa-amylase (sAA), salivary cortisol (sCor), salivary immunoglobulin A (sIgA) and salivary dehydroepiandrosterone (sDHEA) have been regarded as reasonable markers [24,25,26,27]. As a response to psychological stress, sAA, sCor, and sDHEA increase and sIgA decreases [24,25,26,27,28,29,30]. Recent research also supports the use of heart rate variability (HRV) as an objective marker in assessing the level of psychological stress [31]. HRV is defined as the fluctuation of the length of heartbeat intervals, and it can be utilized to indirectly evaluate the changes in the activity of the autonomic nervous system (ANS). Increased work stress has been reported to be associated with a higher heart rate [32]. A higher level of stress has also been associated with lower HRV during an orthostatic test [33].

In terms of personal sleep deprivation and poor sleep quality, high scores on the Epworth Sleepiness Scale (ESS) and Pittsburgh Sleep Quality Index (PSQI) have been associated with degraded psychomotor vigilance performance in a naval environment [34,35]. Overall, moderate physical exercise is considered to be beneficial regarding sleep quality [36]. Considering shift work, older workers have been discovered to face more sleep disturbances than younger ones [37]. Older age has also been associated with increased sleep problems, in general [38,39]. Concerning shift work and especially older workers, a very rapidly forward-rotating shift system has been found to positively affect sleep when compared to a slower backwards-rotating system [40]. In a naval environment, during more irregular working hours, fixed watchkeeping systems have been considered more beneficial regarding fatigue than rotating systems [41]. When assessing the amount of subjective sleepiness, the Karolinska Sleepiness Scale (KSS) has been considered a valid marker and it may also represent a potential marker regarding fatigue [42,43,44,45]. When assessing cognitive capability, the Sustained Attention to Response Task (SART) has been used to evaluate sustained attention and inhibitory control [46], whereas the N-Back Task (N-Back) has been considered a useful test regarding working memory [47].

The aim of this study was to investigate individual characteristics of the naval personnel that have associations with the amount of sleepiness, fatigue and stress responses during shift work and irregular working hours in a naval environment. The examined individual characteristics of the participants were: their age, prior time in naval service, subjective level of physical activity, body composition, physical fitness, blood biomarkers and psychological factors. It was hypothesized that greater physical activity and a higher level of physical fitness in addition to a healthy lifestyle and strong mental performance could attenuate stress responses on board.

## 2. Materials and Methods

### 2.1. Study Design on Board

The onboard measurements took place on a Finnish Defence Forces’ (FDF) Navy missile patrol boat and consisted of two separate study periods. The measurements lasted 2 weeks in total: 1 week (7 days and 6 nights) with a 4:4 watchkeeping system and another week (7 days and 6 nights) with a 4:4/6:6 watchkeeping system. The structures of the watchkeeping systems are shown in Figure 1 and Figure 2. Both watchkeeping systems consisted of two watch sections and, therefore, required two working groups to maintain the systems. When one working group was working, the second working group was resting and the other way around. This meant that both working groups were working a total of 12 h each day. The 4:4 watchkeeping system contained two daily 2 h half-watches (16:00–18:00 h and 18:00–20:00 h) and had a rotating watch schedule. Because of the rotating watch schedule, the working hours during this 4:4 watchkeeping system were completely irregular between the consecutive days. The 4:4/6:6 watchkeeping system contained the same two daily half-watches (16:00–18:00 h and 18:00–20:00 h) and was a fixed system. Because of the fixed watch schedule, the working periods were at the same time each day. Most of the participants were the same in both study periods, and there were 16 days between the periods to make sure that the participants had sufficient time to recover from the first study week.

The standing watch duties of the study participants consisted of tasks that did not require any notable physical effort. The state of the sea was similar in both study periods, during the first period mean wind speed was 5.6 m/s (range 0–12 m/s) and during the second period, it was 7.4 m/s (range 2–12 m/s). During the first study period, the missile patrol boat monitored and secured Finland’s territorial integrity and during the second study period it operated in a naval exercise. The onboard measurements consisted of the Karolinska Sleepiness Scale (KSS), salivary alfa-amylase (sAA), salivary cortisol (sCor), salivary immunoglobulin A (sIgA), salivary dehydroepiandrosterone (sDHEA), cognitive tests (SART and N-Back) and heart rate variability (HRV) during an orthostatic test.

### 2.2. Participant Characteristics

In total, 18 healthy male FDF Navy soldiers (n = 14) and conscripts (n = 4) took part in the study. During the first study week (4:4 watchkeeping system), 17 participants (13 FDF Navy soldiers and 4 conscripts) were studied, while during the second week (4:4/6:6 watchkeeping system), 16 participants (13 FDF Navy soldiers and 3 conscripts) were studied. Fifteen participants were the same in both study periods (12 FDF Navy soldiers and 3 conscripts).

The participants were recruited from the same FDF Navy missile patrol boat class where the measurements were conducted. They gave their informed consent for the study, participated voluntarily and did not receive any financial gain for their participation in the study. The participants were advised to maintain a regular sleep–wake rhythm and avoid sleep deprivation 3 days prior to the measurement periods. They also reported an approximation of the mean duration of their sleep for the last 3 days and the actual duration of their sleep on the last day before the study periods. The FDF Navy soldiers that work in naval duty must be clinically examined every second year before the age of 40 years and every year after the age of 40. Additionally, the conscripts must be clinically examined before naval duty. The examination is performed by a military physician that is an approved medical examiner for seafarers. During this clinical examination, the soldiers are also screened for sleep apnea and should not have any notable sleep problems. All participants that were clinically fit to work in a naval environment were included in the study.

The participants’ age, prior time in naval service and subjective level of physical activity were determined at the beginning of the study periods. The body composition, physical fitness and blood biomarkers were determined within 3 months of the onboard study periods. The psychological factors of the participants were determined within 1 year of the onboard study periods.

Age and prior time in the naval service were defined in years. The level of subjective physical activity was defined as a numeric value from 0 to 10 according to the physical activity classification shown in Figure 3. Regarding the body composition, the body mass index (BMI), body fat percentage (BFP), skeletal muscle mass (SMM) and fat mass (FATM) were measured in the morning after 10 h of fasting using a segmental multifrequency bioimpedance analysis (InBody 720, Biospace, Seoul, South Korea). The waist circumference (WC) of the participants was also measured.

Concerning physical fitness, a number of physical fitness tests were used: the maximal number of push-ups and sit-ups in one minute, a standing long jump and a 12-min running test. The participants were familiar with these physical fitness tests because they belong to basic military training in the FDF. In addition, the maximal voluntary contraction of the upper (MVCupper) and lower (MVClower) extremities were measured with an electromechanical dynamometer manufactured by the University of Jyväskylä (Jyväskylä, Finland). Both measurements were conducted bilaterally and in a sitting position. When taking the MVCupper measurements, the handlebar was maintained at the height of the shoulders with a 90° angle from the elbows. During the MVClower measurements, the knee and hip angles were maintained at 107° and 110°. A seated medicine ball throw (SMBT) was also performed to assess the rapid power production of the upper limbs. The ball (2kg) was thrown with both hands, keeping forearms equivalent to the floor, back against a wall and legs straight. Distance from the landing point to the wall was measured. A supervisor observed all tests and also demonstrated the correct techniques for each test prior to conducting the tests.

Venous blood samples were collected from the antecubital vein after an overnight 10-h fast. The following biomarkers were analyzed: the fasting plasma glucose (FPG), fasting plasma insulin (FPI), hemoglobin A1c (HbA1c), total cholesterol (TC), low-density lipoprotein cholesterol (LDL-C), high-density lipoprotein cholesterol (HDL-C), testosterone (TES) and insulin-like growth factor-1 (IGF-1). The IGF-1 samples were analyzed by the Bioanalytical Laboratory Unit of the Faculty of Sport and Health Sciences (Jyväskylä, Finland). All other blood biomarkers were analyzed by the Tykslab Operational Division Laboratory of the Intermunicipal Hospital District of Southwest Finland.

The psychological measures of this study were the Finnish version of the Short Five personality test [48] and the Finnish version of the shortened resilience scale [49]. The Short Five personality test (S5) is a shortened version of the Big Five personality test. It evaluates personality by measuring the five main personality traits: extraversion, neuroticism, openness, agreeableness and conscientiousness [48]. RS14 is a shortened version of the Resilience Scale (RS) and is a valid tool for measuring resilience [49]. A more detailed description of the individual characteristics of the study participants is shown in Table 1.

### 2.3. Measurements on Board the Navy Missile Patrol Boat

#### 2.3.1. The Karolinska Sleepiness Scale (KSS)

During study periods, the participants reported numeric KSS values in individual sleep diaries at the beginning and end of each watch. The scale consisted of values from 1 (extremely alert) to 9 (extremely sleepy or fighting sleep).

#### 2.3.2. Salivary AA, Cor, IgA and DHEA

Saliva samples were collected daily at 16:00 h or 18:00 h depending on the watchkeeping system with a Salivette^®^ sampling device in accordance with the instructions of the device. The participants did not use any cortisol medication during or before the measurements. They were advised to avoid physical exercise for 3 h and to avoid brushing their teeth or eating for 1 h before giving the saliva sample. The collected saliva samples were stored in a freezer before further analysis. After defrosting, sAA, sCor, sIgA and sDHEA were analyzed by the Bioanalytical Laboratory Unit of the Faculty of Sport and Health Sciences (Jyväskylä, Finland).

#### 2.3.3. Orthostatic Test

After the collection of the saliva samples, an orthostatic test, with 5 min supine (SU) and 5 min standing (ST), was performed as a daily measurement point for the HRV measures. All study participants on the same watch section performed the orthostatic test at the same time and were advised to avoid all other physical activities while completing the test.

#### 2.3.4. Cognitive Tests (SART, N-Back)

After completing the orthostatic test, participants performed SART and N-Back with laptop computers. New, unused keyboards were attached to all computers and the participants completed the tests using the same keyboard and computer on all measurements. A more precise conduction of the cognitive tests is described in a prior study [50].

In the present study, the mean reaction times in the correct response trials (SART RT) and the number of commission errors (SART Errors) were evaluated using SART. In the N-Back, the number of correct responses (N-Back Total hits) and the number of commission errors (N-Back Errors) were evaluated.

#### 2.3.5. Heart Rate Variability (HRV)

The HRV was recorded with the Bodyguard 2 device (Firstbeat Technologies Ltd., Jyväskylä, Finland) which recorded R to R intervals at a sampling frequency of 1000 Hz. The HRV during an orthostatic test was recorded daily, except on the fourth day of both study periods because the HRV measurement device had to be recharged. Regarding the analysis, there were 1-min baseline recordings in the SU and ST positions so that, in both positions, only the last 4 min of HRV were used for analysis to obtain reliable results [51]. The analysis was performed using the Kubios HRV Standard program (version 3.4.3, Kubios Ltd., Kuopio, Finland). In the present study, the mean heart rate (HRmean), the standard deviation of NN intervals (SDNN), root mean square of successive RR interval differences (RMSSD), absolute total power (TP), the absolute power of the very low-frequency band (VLF), absolute power of the low-frequency band (LF), absolute power of the high-frequency band (HF), and the ratio of LF to HF power (LF/HF) were investigated.

A recent study assessing stress during an orthostatic test pointed out that a higher level of stress was associated with a lower HRV overall [33]. Still, a frequently reported finding in terms of HRV and stress is low parasympathetic activity, which is perceived as a decrease in HF and an increase in LF power [31]. Despite the fact that in long-term ambulatory recordings the LF power has been reported to increase with sympathetic activity [52], this does not seem to occur in short-term resting recordings [53]. In short-term resting recordings, the LF power has been reported to increase with slower breathing and it is almost an entirely vagally mediated parasympathetic activity [53]. A high LF/HF ratio has also been considered a controversial marker for sympathetic activity and should be used with caution in short-term recordings [53]. In the present study, an elevated HRmean, lower HRV overall and higher LF/HF ratio were considered stress responses during the orthostatic test.

### 2.4. Statistics

Statistical analysis was performed using the SPSS statistical software (SPSS version 27.0.1.0; SPSS Inc., Chicago, IL, USA). A Shapiro–Wilk test was used to assess the normal distribution of the data. Correlations between normally distributed variables were calculated using a Pearson correlation coefficient, and a Spearman correlation coefficient was used for non-normally distributed variables. Correlations between the individual characteristics and KSS were calculated using the mean or median of absolute KSS values. All other correlations between the individual characteristics and onboard measurements were conducted using the mean or median of absolute differences between the baseline values (first measured values) and follow-up values (later measured values) of the onboard measurements. Any *p*-values lower than 0.05 were considered statistically significant.

## 3. Results

### 3.1. The Most Distinct Associations between Sleepiness, Fatigue, Stress Responses and Individual Characteristics

The participants’ age and prior time in naval service had the most distinct positive associations with the amount of sleepiness and fatigue and the stress responses during the study weeks. In contrast, their physical activity classification and standing long jump had negative associations. Regarding sleepiness and fatigue, the correlations between these individual characteristics and KSS scores overall are displayed in Figure 4. Significant positive associations with KSS scores overall during both study weeks were observed regarding age (1. r = 0.52, *p* = 0.03 and 2. r = 0.58, *p* = 0.02) and prior time in the naval service (1. r = 0.74, *p* < 0.01 and 2. r = 0.73, *p* < 0.01), whereas significant negative associations with KSS scores overall were observed regarding physical activity (1. r = −0.61, *p* < 0.01 and 2. r = −0.55, *p* = 0.03) and standing long jump (2. r = −0.54, *p* = 0.03).

A description of statistically significant correlations between the most distinct individual characteristics of the study participants and their sleepiness, fatigue and stress responses during the study weeks is shown in Table 2. Higher age was also associated with higher KSS scores during the days at 16:00–18:00 h and KSS scores during the nights at 02:00–04:00 h. Concerning stress response, higher age was associated with higher levels of sDHEA and lower levels of LF (SU). On the other hand, it was also associated with lower levels of sAA. A longer prior time in the naval service was positively associated with KSS scores during the days at 16:00–18:00 h and KSS scores during the nights at 02:00–04:00 h. Regarding stress response, a longer prior time in naval service was associated with higher levels of sDHEA and LF/HF (SU) and lower levels of TP (ST) and SDNN (ST). In contrast, higher physical activity of the study participants was associated with lower KSS scores during the days at 16:00–18:00 h, KSS scores during the nights at 02:00–04:00 h, and lower levels of LF/HF (SU). Higher results in the standing long jump were also associated with lower KSS scores during the days at 16:00–18:00 h. Regarding stress response, higher results in the standing long jump were associated with lower levels of the HRmean (ST) and higher levels of SDNN (ST) and RMSSD (ST).

### 3.2. Associations regarding Physical Fitness and Body Composition

A description of statistically significant correlations concerning physical fitness and body composition is shown in Table 3. Regarding physical fitness, in addition to the prior mentioned physical classification and standing long jump, push-ups and the MVCupper test values had negative associations with stress responses in terms of HRV or salivary stress hormones. Higher numbers of push-ups were associated with lower levels of sCOR, and higher MVCupper results were associated with lower levels of sAA and LF/HF (SU). Sit-ups, MVClower, SMBT and the 12-min run test had positive associations with the level of cognitive performance during the study weeks. They were all associated with faster reaction times in the SART, while MVClower was also associated with a smaller number of commission errors in the SART. On the other hand, the 12-min run test also had a positive association with the stress responses when it was associated with a lower LF (ST).

Regarding body composition, BMI, WC, BFP and FATM had mainly positive associations with the stress responses in HRV. They were all associated with lower TP (SU). On the other hand, FATM was also positively associated with TP (SU) in the second study period. The WC, BFP and FATM were also negatively associated with LF (SU). Concerning the cognitive tests, higher WC was associated with a smaller number of total hits in the N-Back test. Higher SMM was associated with a higher number of total hits in the N-Back test, but in terms of stress responses, it was also associated with higher levels of sAA.

### 3.3. Associations regarding Blood Biomarkers

In terms of blood biomarkers, statistically significant correlations are shown in Table 4. FPG, FPI, HbA1c, TC and LDL-C, had positive associations and TES, HDL-C and IGF-1 had negative associations with the stress responses in HRV. FPG was associated with lower LF (SU), TP (SU), SDNN (ST) and RMSSD (ST). FPI was associated with lower RMSSD (SU), HF (SU) and TP (SU). HbA1c was associated with higher LF/HF (SU). TC and LDL-C were associated with lower LF (ST), and LDL-C was also associated with higher LF/HF (ST). In contrast, TES was associated with higher SDNN (SU), RMSSD (SU), HF (SU) and with lower LF/HF (ST). HDL-C was associated with lower HRmean (ST) and higher LF (ST). IGF-1 was associated with lower LF/HF (SU).

Higher FPG was also associated with higher sCOR and FPI with higher sAA. Regarding the cognitive tests, higher values of TES and HDL-C were associated with a smaller number of commission errors in SART and with faster SART reaction times. Higher HDL-C was also associated with a greater number of total hits in the N-Back test. In contrast, higher FPG was associated with a smaller number of total hits in the N-Back test.

### 3.4. Associations regarding Psychological Factors

In terms of psychological factors, statistically significant correlations are displayed in Table 5. In S5 extraversion was associated with lower KSS scores during days at 16:00–18:00 h. Neuroticism was associated with higher levels of HRV. It was associated with higher SDNN (SU), RMSSD (SU) and LF (SU). Openness was associated with faster reaction time in SART and lower levels of LF/HF (ST). Agreeableness was associated with higher levels of LF (SU) and lower sCOR, but it was also associated with a higher number of errors in N-Back. Higher RS14 results were associated with higher levels of LF/HF (ST).

## 4. Discussion

The aim of this study was to investigate individual characteristics of the naval personnel that have associations with the amount of sleepiness, fatigue and stress responses during shift work and irregular working hours. In the present study, older participants and those who had spent more time in the naval service were most distinctly associated with higher levels of sleepiness, fatigue and stress on board. In contrast, greater amounts of physical activity and a higher level of physical fitness, especially standing long jump, were most distinctly associated with lower levels of sleepiness, fatigue and fewer stress responses. In addition, higher BMI, BFP, FATM, broader WC and higher levels of FPG, FPI, HbA1c, TC and LDL-C seemed to be associated with a higher level of stress, whereas higher levels of TES, HDL-C and IGF-1 seemed to be associated with fewer stress responses.

In terms of age, the present study results are well in line with prior shift work studies, where it has been found that older workers face more sleep disturbances and more difficulties in adjusting to consecutive night shifts than younger workers [37,40]. Older age has also been associated with problems with the adjustment of circadian rhythms and it has been associated with increased sleep problems, in general [38,39]. According to the present study, previous time in the naval service was not a protective factor considering age and appeared to be clearly associated with an increased amount of sleepiness, fatigue and stress in naval service.

Physical activity and the level of physical fitness, especially better performance in the standing long jump, seemed to reduce sleepiness, fatigue and stress responses in the present study. In the current literature, physical activity is mainly considered to be associated with less subjective psychological stress and a better ability to cope with stressful situations [7]. In recent studies, greater levels of cardiorespiratory and muscular fitness have also been associated with lower stress symptoms among normal-weight men [5,6]. Moderate physical exercise is also known to have benefits regarding the quality of sleep in all age groups [36]. A physical training intervention can also be a useful tool for shift workers that work irregular shifts to reduce work-dependent fatigue [54].

Regarding body composition, the BMI, WC, BFP and FATM seemed to increase stress responses according to HRV in the present study. This is in line with prior studies, where especially a higher BMI and BFP have been associated with increased levels of stress [8,9]. It has also been pointed out that increased stress responses due to higher BFP may impair cognitive performance [9]. In the present study, similar effects were discovered, as WC was negatively and SMM was positively associated with cognitive performance in the N-Back test.

In terms of blood biomarkers, FPG, FPI, HbA1c, TC and LDL-C seemed to increase stress responses and TES, HDL-C and IGF-1 seemed to reduce stress responses according to the HRV. The current literature also points out that a high muscle mass together with a lower fat mass may reduce insulin resistance [55], and that, in soldiers, a greater amount of body fat has been associated with higher TC and LDL-C [56]. Prior studies have also pointed out that TES appears to stimulate IGF-1 expression, leading to increased muscle growth and it may also increase lipolysis and reduce adipogenesis [57]. Therefore, the present associations between the laboratory values and the stress responses seem to be consistent with the results regarding body composition and stress responses. In addition, HDL-C seemed particularly associated with better cognitive performance during the study periods, and in prior studies, it has also been pointed out that HDL-C levels may be positively associated with cognitive performance [58]. The present study suggests that an athletic body composition together with a healthy lifestyle may be beneficial concerning stress responses on board.

Regarding psychological factors, even though the results were quite inconclusive overall, in the S5 extraversion seemed to be associated with a reduced amount of sleepiness. This finding is in line with prior studies, in which extraversion has been positively linked with shift work tolerance [22]. On the other hand, in the present study neuroticism seemed to be associated with lower stress responses regarding HRV. This finding is not in line with prior studies, in which neuroticism has been negatively linked with shift work tolerance [22]. In the present study, the scores regarding neuroticism of the participants were very low which points out that there was no indication that high scores of neuroticism would be associated with lower stress responses. The low neuroticism scores may be explained by the selection process of the naval soldiers and by the fact that people with less neuroticism seem more likely to enter the military [59]. In the present study, openness seemed to be associated with better cognitive performance and lower stress responses regarding HRV. This finding is in line with the previous literature, which finds that openness could be a protective factor against burnout [23].

To our knowledge, the present study is the first to provide information regarding the individual characteristics that might have an impact on the amount of sleepiness, fatigue and stress responses during shift work and irregular working hours in a maritime environment. The use of multiple measures on board also makes it quite unique compared to previous studies regarding the naval environment and provides deeper knowledge on the stress responses during patrolling at sea. In the present study, the same measurements were also completed in two separate study periods, meaning that the study design was repeated once during the study. Considering the different characteristics of the study participants, the measures were quite consistent with each other in terms of beneficial and non-beneficial associations. The present study results are also well in line with the previous literature regarding shift work and irregular working hours.

The most significant limitation of the present study is the relatively small number of study participants, and some statistically non-significant associations could be simply due to this small sample size. In addition, some of the statistically significant associations could also be due to only a few sedentary participants. There can also be other confounding factors considering the study participants, which might mask an actual association or falsely demonstrate an apparent association. The study was based on participants who worked in a certain vessel class during certain weather conditions, as there were no major differences in wind conditions or the state of the sea between the study periods. For further research, additional studies which include larger samples, more ship departments, longer measurement periods and the elimination of possible confounding factors are needed.

## 5. Conclusions

The present study suggests that older age and a longer time in naval service are most distinctly associated with a greater amount of sleepiness, fatigue and stress during shift work and irregular working hours in a naval environment. In contrast, higher amounts of physical activity and a higher level of physical fitness, especially standing long jump, seem to be most distinctly associated with less sleepiness, fatigue and stress. In addition, higher BMI; BFP; FATM; broader WC; and higher levels of FPG, FPI, HbA1c, TC and LDL-C appear to be associated with higher stress responses, whereas higher levels of TES, HDL-C and IGF-1 appear to be associated with lower stress responses. From the psychological perspective, extraversion and openness may be beneficial factors in terms of sleepiness and stress responses on board. The results highlight the importance of physical activity, physical fitness and a healthy lifestyle during service in a naval environment.

## Figures and Tables

**Figure 1 ijerph-19-13451-f001:**
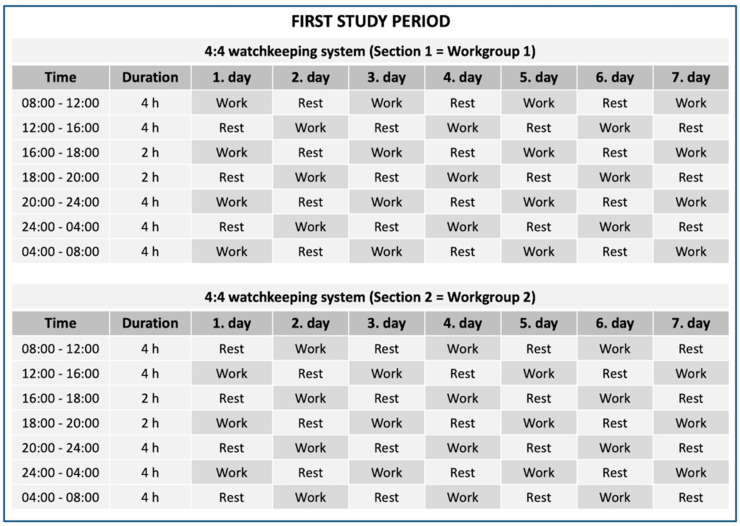
The structure of the two Section 4:4 watchkeeping system during the first study period.

**Figure 2 ijerph-19-13451-f002:**
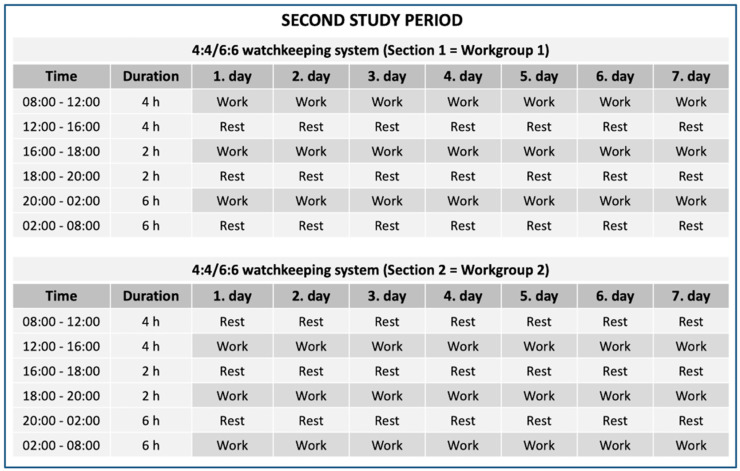
The structure of the two Section 4:4/6:6 watchkeeping system during the second study period.

**Figure 3 ijerph-19-13451-f003:**
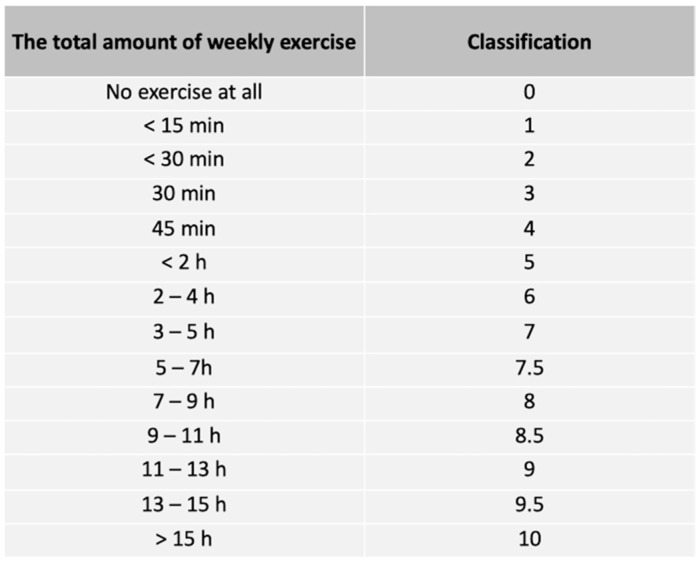
The physical activity classification.

**Figure 4 ijerph-19-13451-f004:**
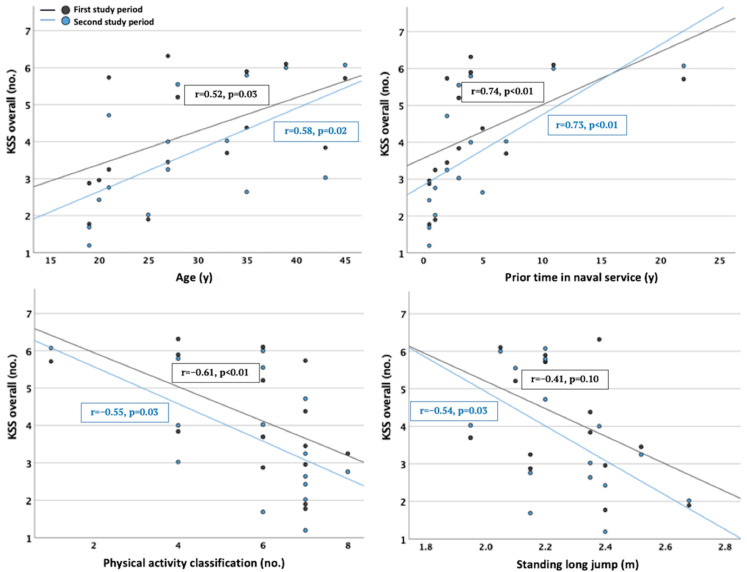
The correlations between the most distinct individual characteristics (age, prior time in naval service, physical activity and standing long jump) and mean Karolinska Sleepiness Scale (KSS) scores overall.

**Table 1 ijerph-19-13451-t001:** Description of the individual characteristics of the study participants. Range, mean and standard deviation (SD) are presented.

	FIRST STUDY PERIOD			SECOND STUDY PERIOD		
	n	Range	Mean (SD)	n	Range	Mean (SD)
Age (y)	17	19–45	29 (8)	16	19–45	29 (9)
Prior time in naval service (y)	17	0.5–22.0	4.2 (5.4)	16	0.5–22.0	4.8 (5.6)
Physical activity classification (no.)	17	1–8	6 (2)	16	1–8	6 (2)
**Body composition**						
Height (m)	16	1.65–1.87	1.79 (0.05)	15	1.65–1.87	1.80 (0.05)
Body mass (kg)	16	62.8–102.1	82.8 (8.7)	15	62.8–102.1	83.8 (8.4)
Body mass index (kg/m^2^)	16	19.9–32.0	25.9 (3.0)	15	19.9–32.0	25.9 (3.0)
Body fat percentage (%)	16	7.7–34.4	20.0 (6.8)	15	7.7–34.4	20.3 (7.0)
Skeletal muscle mass (kg)	16	32.8–42.8	37.5 (3.1)	15	32.8–42.8	37.7 (3.0)
Fat mass (kg)	16	4.8–31.2	16.9 (6.9)	15	4.8–31.2	17.4 (7.1)
Waist circumference (cm)	11	67–113	91 (12)	12	67–113	91 (11)
**Physical fitness**						
Push-ups (reps/min)	17	22–68	41 (11)	16	22–68	40 (11)
Sit-ups (reps/min)	17	22–62	45 (10)	16	22–62	44 (10)
Standing long jump (m)	17	1.95–2.68	2.29 (0.19)	16	1.95–2.68	2.29 (0.19)
12-min run test (m)	17	2030–3010	2628 (293)	16	2030–3010	2634 (277)
MVCupper (kg)	11	84–124	106 (11)	12	81–124	104 (12)
MVClower (kg)	11	323–531	444 (59)	12	292–531	432 (71)
SMBT (m)	11	5.10–8.02	6.57 (0.80)	12	5.10–8.02	6.50 (0.80)
**Blood biomarkers**						
FPG (mmol/l)	11	5.2–6.1	5.5 (0.2)	12	5.0–6.1	5.5 (0.3)
FPI (mU/l)	11	5.0–17.0	8.3 (4.0)	12	5.0–17.0	8.5 (3.9)
HbA1c (mmol/mol)	11	29.0–36.0	31.9 (2.0)	12	29.0–36.0	32.2 (2.0)
TC (mmol/l)	11	3.2–6.3	4.8 (0.9)	12	3.2–6.3	4.8 (0.9)
LDL-C (mmol/l)	11	1.2–4.6	3.0 (0.9)	12	1.2–4.6	3.0 (0.9)
HDL-C (mmol/l)	11	1.1–2.0	1.5 (0.3)	12	1.1–2.0	1.5 (0.3)
TES (nmol/l)	11	8.7–24.0	15.7 (4.8)	12	8.7–24.0	15.4 (4.7)
IGF-1 (nmol/l)	11	11.2–23.8	18.0 (4.2)	12	11.2–23.8	18.2 (4.1)
**Psychological factors**						
S5: Extraversion (no.)	11	−12–12	3 (10)	11	−21–12	1 (11)
S5: Neuroticism (no.)	11	−31–3	−17 (10)	11	−29–3	−15 (9)
S5: Openness (no.)	11	−6–31	10 (12)	11	−6–31	11 (11)
S5: Agreeableness (no.)	11	−4–26	12 (10)	11	−4–26	11 (10)
S5: Conscientiousness (no.)	11	4–32	19 (9)	11	4–27	17 (8)
RS14 (no.)	11	62–95	79 (11)	11	62–94	77 (9)
**Before measurement periods**						
Sleep/day, last 3 days (h)	16	5.5–8.0	7.0 (0.7)	15	6.0–8.8	7.5 (0.7)
Sleep/day, last 1 day (h)	16	6.5–9.0	7.4 (0.7)	15	5.0–10.0	7.0 (0.8)

Note. MVCupper = maximal voluntary contraction of the upper extremities, MVClower = maximal voluntary contraction of the lower extremities, SMBT = seated medicine ball throw, FPG = fasting plasma glucose, FPI = fasting plasma insulin, HbA1c = hemoglobin A1c, TC = total cholesterol, LDL-C = low-density lipoprotein cholesterol, HDL-C = high-density lipoprotein cholesterol, TES = testosterone, IGF-1 = insulin-like growth factor-1, S5 = the Short Five Personality test, RS14 = shortened version of the Resilience Scale.

**Table 2 ijerph-19-13451-t002:** Statistically significant correlations regarding the most distinct individual characteristics.

	FIRST STUDY PERIOD			SECOND STUDY PERIOD	
	KSS	sAA, sCOR, sIgA, sDHEA	HRV	KSS	HRV
Age	•KSS (overall) r = 0.52 * •KSS (02–04 h) r = 0.54 *	•sDHEA r = 0.55 * °sAA r = −0.52 *	•LF(SU) r = −0.59 *	•KSS (overall) r = 0.58 * •KSS (16–18 h) r = 0.58 * •KSS (02–04 h) r = 0.50 *	-
Prior time in naval service	•KSS (overall) r = 0.74 ** •KSS (16–18 h) r = 0.71 ** •KSS (02–04 h) r = 0.70 **	•sDHEA r = 0.61 **	•SDNN(ST) r = −0.60 * •TP(ST) r = −0.55 *	•KSS (overall) r = 0.73 ** •KSS (16–18 h) r = 0.67 ** •KSS (02–04 h) r = 0.65 **	•LF/FH(SU) r = 0.56 *
Physical activity classification	°KSS (overall) r = −0.61 ** °KSS (16–18 h) r = −0.56 * °KSS (02–04 h) r = −0.61 **	-	-	°KSS (overall) r = −0.55 * °KSS (16–18 h) r = −0.54 *	°LF/HF(SU) r = −0.69 **
Standing long jump	°KSS (16–18 h) r = −0.48 *	-	°SDNN(ST) r = 0.55 *	°KSS (overall) r = −0.54 * °KSS (16–18 h) r = −0.61 *	°HRmean(ST) r = −0.55 * °RMSSD(ST) r = 0.65 **

Note. KSS = Karolinska Sleepiness Scale, sAA = salivary alfa-amylase, sCor = salivary cortisol, sIgA = salivary immunoglobulin A, sDHEA = salivary dehydroepiandrosterone, HRV = heart rate variability (SU = supine, ST = standing, HRmean = mean heart rate, SDNN = standard deviation of NN intervals, RMSSD = the root mean square of successive RR interval differences, TP = absolute total power, LF = absolute power of the low frequency band, LF/HF = the ratio of LF to HF power). * Significance *p* < 0.05; ** *p* < 0.01. ° = beneficial association and • = non-beneficial association considering sleepiness, fatigue or stress responses on board.

**Table 3 ijerph-19-13451-t003:** Statistically significant correlations regarding physical fitness and body composition.

	FIRST STUDY PERIOD			SECOND STUDY PERIOD		
	sAA, sCOR, sIgA, sDHEA	SART, N-Back	HRV	sAA, sCOR, sIgA, sDHEA	SART, N-Back	HRV
Push-ups	-	-	-	°sCOR r = −0.55 *	-	-
Sit-ups	-	-	-	-	°SART RT r = −0.52 *	-
MVCupper	°sAA r = −0.75 **	-	-	-	-	°LF/HF (SU) r = −0.67 *
MVClower	-	°SART Errors r = −0.63 * °SART RT r = −0.65 *	-	-	-	-
SMBT	-	-	-	-	°SART RT r = −0.63 *	-
12-min run test	-	-	•LF (ST) r = −0.64 *	-	°SART RT r = −0.56 *	-
BMI	-	-	•TP (SU) r = −0.62 *	-	-	-
WC	-	•N-Back Total hits r = −0.66 *	•LF (SU) r = −0.77 ** •TP (SU) r = −0.83 **	-	-	-
BFP	-	-	•LF (SU) r = −0.78 ** •TP (SU) r = −0.75 **	-	-	-
FATM	-	-	•LF (SU) r = −0.81 ** •TP (SU) r = −0.74 **	-	-	°TP (SU) r = 0.55 *
SMM	-	-	-	•sAA r = 0.63 *	°N-Back Total hits r = 0.56 *	-

Note. sAA = salivary alfa-amylase, sCor = salivary cortisol, sIgA = salivary immunoglobulin A, sDHEA = salivary dehydroepiandrosterone, SART = the Sustained Attention to Response Task (SART RT = the mean reaction time in the correct response trials, SART Errors = the number of commission errors), N-Back = the N-back Task (N-Back Total hits = the number of correct responses), HRV = heart rate variability (SU = supine, ST = standing, TP = absolute total power, LF = absolute power of the low frequency band, LF/HF = the ratio of LF to HF power), MVCupper = maximal voluntary contraction of the upper extremities, MVClower = maximal voluntary contraction of the lower extremities, SMBT = seated medicine ball throw, BMI = body mass index, WC = waist circumference, BFP = body fat percentage, FATM = fat mass, SMM = skeletal muscle mass. * Significance *p* < 0.05; ** *p* < 0.01. ° = beneficial association and • = non-beneficial association considering sleepiness, fatigue or stress responses on board.

**Table 4 ijerph-19-13451-t004:** Statistically significant correlations regarding blood biomarkers.

	FIRST STUDY PERIOD			SECOND STUDY PERIOD		
	sAA, sCOR, sIgA, sDHEA	SART, N-Back	HRV	sAA, sCOR, sIgA, sDHEA	SART, N-Back	HRV
FPG	-	•N-Back Total hits r = −0.63 *	•LF (SU) r = −0.80 ** •TP (SU) r = −0.91 ** •SDNN (ST) r = −0.72 * •RMSSD (ST) r = −0.71 *	•sCOR r = 0.59 *	-	-
FPI	•sAA r = 0.71 *	-	•RMSSD (SU) r = −0.68 * •HF (SU) r = −0.73 * •TP (SU) r = −0.75 *	-	-	-
HbA1c	-	-	-	-	-	•LF/HF (SU) r = 0.58 *
TC	-	-	-	-	-	•LF (ST) r = −0.67 *
LDL-C	-	-	-	-	-	•LF (ST) r = −0.59 * •LF/HF (ST) r = 0.68 *
HDL-C	-	°SART Errors r = −0.66 * °SART RT r = −0.64 *	-	-	°N-Back Total hits r = 0.60 *	°HRmean (ST) r = −0.63 * °LF (ST) r = 0.61 *
TES	-	°SART Errors r = −0.81 ** °SART RT r = −0.77 **	°HF (SU) r = 0.64 * °LF/HF (ST) r= −0.78 *	-	-	°SDNN (SU) r = 0.67 * °RMSSD ( SU) r = 0.65 *
IGF-1	-	-	-	-	-	°LF/HF (SU) r = −0.69 *

Note. sAA = salivary alfa-amylase, sCor = salivary cortisol, sIgA = salivary immunoglobulin A, sDHEA = salivary dehydroepiandrosterone, SART = the Sustained Attention to Response Task (SART RT = the mean reaction time in the correct response trials, SART Errors = the number of commission errors), N-Back = the N-back Task (N-Back Total hits = the number of correct responses), HRV = heart rate variability (SU = supine, ST = standing, HRmean = mean heart rate, SDNN = standard deviation of NN intervals, RMSSD = the root mean square of successive RR interval differences, TP = absolute total power, LF = absolute power of the low frequency band, HF = absolute power of the high frequency band, LF/HF = the ratio of LF to HF power), FPG = fasting plasma glucose, FPI = fasting plasma insulin, HbA1c = hemoglobin A1c, TC = total cholesterol, LDL-C = low-density lipoprotein cholesterol, HDL-C = high-density lipoprotein cholesterol, TES = testosterone, IGF-1 = insulin-like growth factor-1. * Significance *p* < 0.05; ** *p* < 0.01. ° = beneficial association and • = non-beneficial association considering sleepiness, fatigue or stress responses on board.

**Table 5 ijerph-19-13451-t005:** Statistically significant correlations regarding psychological factors.

	FIRST STUDY PERIOD		SECOND STUDY PERIOD		
	KSS	HRV	sAA, sCOR, sIgA, sDHEA	SART, N-Back	HRV
S5: Extraversion	°KSS (16–18 h) r = −0.60 *	-	-	-	-
S5: Neuroticism	-	-	-	-	°SDNN(SU) r = 0.87 ** °RMSSD(SU) r = 0.87 ** °LF(SU) r = 0.68 *
S5: Openness	-	-	-	°SART RT r = −0.64 *	°LF/HF(ST) r = −0.68 *
S5: Agreeableness	-	°LF (SU) r = 0.72 *	°sCOR r = −0.62 *	•N-Back Errors r = 0.61 *	-
RS14	-	•LF/FH (SU) r = 0.68 *	-	-	-

Note. KSS = Karolinska Sleepiness Scale, sAA = salivary alfa-amylase, sCor = salivary cortisol, sIgA = salivary immunoglobulin A, sDHEA = salivary dehydroepiandrosterone, SART = the Sustained Attention to Response Task (SART RT = the mean reaction time in the correct response trials), N-Back = the N-back Task (N-Back Errors = the number of commission errors), HRV = heart rate variability (SU = supine, ST = standing, SDNN = standard deviation of NN intervals, RMSSD = the root mean square of successive RR interval differences, LF = absolute power of the low frequency band, LF/HF = the ratio of LF to HF power), S5 = the Short Five Personality test, RS14 = shortened version of the Resilience Scale. * Significance *p* < 0.05; ** *p* < 0.01. ° = beneficial association and • = non-beneficial association considering sleepiness, fatigue or stress responses on board.

## Data Availability

The datasets used and analyzed during the current study are the property of the Navy Command Finland. All data is primarily not public, but the data availability can be sought from the corresponding author on reasonable request.

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
