# Peer review of "The Effects of Individual Characteristics of the Naval Personnel on Sleepiness and Stress during Two Different Watchkeeping Schedules"

_ijerph, 2022, doi:10.3390/ijerph192013451_

Round 1
Reviewer 1 Report
Thank you for your interesting paper.
Please mention the ethical approval for the research in the paper.
Please complete the introduction part with relevant information/studies related to the aspects investigated in your paper: sleepiness and fatigue, blood biomarkers, heart rate, psychological factors involved.
Author Response
Please see the the attachment.

Reviewer 2 Report
Referee comments
This a rare field study of navy seafarers aiming to investigate the effect of individual differences of the personnel on sleepiness, fatigue, and stress responses during the service in a true environmental. The sample is based on 18 crew members, with 15 being the same during the two naval patrol boat trips of about one week each. The two trips were done with different shift systems, the effects of which on sleepiness have been reported earlier (Chronobiology International). The individual differences included extensive information on individual differences associated to basic demographics, physical fitness, some biomarkers and psychological tests. Outcomes were subjective sleepiness (KSS), some stress hormones, hear rate variation and cognitive tests.
Major comments
The methodology of the study are extensive, but the results stay partly inconclusive due to the small number of subjects for a study on individual differences (n=18). The reporting is also focused on statistically significant results while many non-significant associations can be simply due to the too small sample. Some of the main results could also be due to only few outliners. Figure 4 shows that some of the main results can easily be due to only two subjects having the lowest level of physical activity and long prior time in naval service.
Even the individual differences have been studies in detail, it seems that the health of the subjects has not been investigated clinically, especially the exclusion of some common sleep disorders like sleep apnea could easily explain the observed association in relation to sleepiness. Were any questionnaires on snoring/sleep apnea used to exclude employees with sleep apnea? Was there any screening or inclusion/exclusion criteria for the subjects?
The MS includes lots of results, it actually reports the effects of the two work schedules (that are called “study periods”?) on many factors (Table 2) by reporting the statistical differences between the baseline and the trip. This was not an aim of the study I understood. The first three chapters of the discussion are not focused on the main results of this study at all, but the results already published (Chronob International) or the effects of the shift schedules on stress etc. (not aims of the present study). The discussion should start from the main results, including both the positive and negative results (now the significant positive associations are mostly highlighted).
In the limitations of the paper, the effect of the small study sample (n= 18) should be discussed critically. Could the selection of the study samples (by navy) influence the results – please discuss
The role of age on the adaptation to shift is discussed superficially, not considering e.g. the specific requirements of the studied two schedules (that were even different: were they rapidly or slowly rotating?), that probably has an influence on the association of age with the adjustment (see e.g. Härmä et al. 2005, Psychophysiology). It is generally assumed that circadian adjustment to permanent nights if more difficult to older employees due to e.g. poorer sleep during the day, but the difference may be different for rapidly or irregular shifts. Also, some early papers on the effects of physical fitness intervention on sleepiness and psychosomatic symptoms (an RCT study) could have been considered in the discussion (Härmä et al. Ergonomics 1988).
The present study also does not include individual differences in sleepiness or sleep length, that could have given more light on the possible reasons why ageing and physical fitness (both influence sleep) had association on sleepiness
Minor comments
Figure 1 does not give an overview of the individual shift schedule during the 10-day trips. What is Section 1, section 2? (are they work/rest periods). In your published paper you had two groups – what happened to them?
Line 201. Please give more information when KSS was measured (at the beginning and end of each watch?)
209 why baseline and follow-up were compared extensively if the aim was to study individual differences?
How the data of 15 same subjects during the two trips were handled in the statistics – hopefully not as independent observations since this was partly a repeated design
456 this may also be due to the small study size. Often studies on individual differences have hundreds or thousands of employees and a prospective design to exclude selection
Round 2
Reviewer 2 Report
The paper has been revised well. Thank you!
Author Response
Thank you!